# A delay in sampling information from temporally autocorrelated visual stimuli

Chloe Callahan-Flintoft [1✉], Alex O. Holcombe [2] & Brad Wyble [3]

Much of our world changes smoothly in time, yet the allocation of attention is typically studied with sudden changes – transients. A sizeable lag in selecting feature information is seen when stimuli change smoothly. Yet this lag is not seen with temporally uncorrelated rapid serial visual presentation (RSVP) stimuli. This suggests that temporal autocorrelation of a feature paradoxically increases the latency at which information is sampled. To test this, participants are asked to report the color of a disk when a cue was presented. There is an increase in selection latency when the disk's color changed smoothly compared to randomly. This increase is due to the smooth color change presented after the cue rather than extrapolated predictions based on the color changes presented before. These results support an attentional drag theory, whereby attentional engagement is prolonged when features change smoothly. A computational model provides insights into the potential underlying neural mechanisms.

[1] Buidling 459 (RDRL-HRS) CCDC Army Research Lab, Aberdeen Proving Ground, MD 21005, USA. [2] Brennan MacCallum Building, School of Psychology, The University of Sydney, Sydney, NSW 2006, Australia. [3] 141 Moore Building, The Pennsylvania State University, University Park, PA 16802, USA. ✉email: ccallahanflintoft@gmail.com

The visual system allows for the selection and prioritization of certain pieces of information in our visual field over others. As our visual input is constantly changing, due to scene changes as well as head and eye movements, the visual system needs the ability to make this selection not only in space but also in time before the information is gone.

Classically, the time course of attention has been studied through the presentation of discrete stimuli either with rapid serial visual presentation (RSVP)[1] or by varying the stimulus onset asynchrony (SOA) between the presentation of the cue and a static target. In the latter studies, the target was immediately masked after it was presented, and the logic was that the minimum SOA needed for accurate performance provides an estimate of the time for attention to arrive at and sample the cued location[2,3]. These studies found that performance rapidly increased as SOA increased to reach a peak ~80–120 ms after the cue. This work has yielded the conventional wisdom for the time needed to shift attention and sample a stimulus. However, it is not clear how straightforwardly such results apply to a continuous stream of potential target stimuli, as is often encountered in real-world scenes.

Extraction of a target at a particular time from a continuous stream of input is a nontrivial problem because the representation of a given stimulus is distributed across cortical areas with distinct processing latencies. Thus, there may be uncertainty regarding the relative time of a cue and the potential target stimuli. Moreover, the visual system may tend to group together stimuli over time, creating a potential segmentation problem, particularly when the features of those stimuli are temporally autocorrelated. For the remainder of this paper the term *smooth* will be used as a term for temporal autocorrelation of stimulus features.

One line of research using smoothly changing features displayed a set of clock faces whose hand smoothly changed throughout the trial. Participants were asked to report the cued clock's hand position at the time of a cue. On average, participants reported the clock hand position presented ~130 ms after the cue[4]. Using a similar clock paradigm, this magnitude of lag has also been seen in tasks exploring divided attention[5] and attentional shifts[6]. It is not limited to the sampling of position. Displaying a disk that changed smoothly in color, Sheth et al.[7] flashed another colored disk on the opposite side of fixation. Participants on average reported the color of the changing disk as it was ~400 ms after the flashed cue disk. Delays were also found for objects changing smoothly in luminance (37 ms), spatial frequency (83 ms), and entropy (95 ms).

Substantial lags for sampling from a stream of stimuli are not always found. Vul et al.[8] and Goodbourn and Holcombe[9] used one or two simultaneous streams of letters and flashed a cuing circle around one of the streams. Participants attempted to report the letter that was presented in that stream at the time of the cue. The average delay estimated was very small, 25 ms or less in both cases. Similarly, Weichselgartner and Sperling[10] found that participants were able to select a letter or digit presented simultaneously with a luminance cue in an RSVP stream. A notable difference with these studies is that those with longer lags used stimuli that changed smoothly over time, while Weichselgartner & Sperling[10], Vul et al.[8], and Goodbourn and Holcombe[9] presented a sequence of unrelated letters or digits that changed abruptly from one to the next. Thus, it may be the case that sharp visual transients influence the latency of attention sampling. Previous studies have found that presenting a transient visual signal at the location of a changing stimulus improves temporal sampling whereas endogenous shifts of attention were not as effective[11].

Based on this difference in findings between these paradigms we propose a theory of attentional drag, which posits that temporal autocorrelation in a visual feature extends the duration of attentional engagement elicited by a cue in order to extract useful information from an object. Thus, attention gets dragged along in time by a temporally autocorrelated feature dimension, which increases the latency to disengage, and results in a delayed feature selection. Conversely, when features change abruptly, as in the case of a conventional RSVP stream of letters, the salient featural change after the cue leads to an earlier disengagement of attention and thereby an earlier selection. Thus, the attentional drag theory proposes that transients can decrease the latency to sample information in part by making it easier for attention to disengage.

By comparing two conditions, one where colors change randomly at a regular interval and another where they change smoothly, Experiment 1 shows empirically that selection can happen immediately in response to a cue in the random condition, but is delayed by ~100 ms in the smooth condition. Critically, in both conditions the presentation rate of the stimuli is the same, the only difference is the temporal autocorrelation of colors. In Experiment 2 these findings are extended to show that increasing the similarity between successive colors further extends the attentional window. Finally, Experiment 3 tested whether these results were the product of feature extrapolation based on the trajectory prior to the cue and find that this explanation is unable to account for the results. Together, the experimental and modeling work presented here demonstrate an influence of featural change in time on information extraction. The neurocomputational model provided in the discussion demonstrates how a simple attentional system can exhibit prolonged selection latencies. In a nutshell, the prolonged latencies reflects a feedback loop between attention-related neurons and sensory neurons. The sensory neurons have overlapping tuning curves that result in greater activation in the case of a smoothly changing stimulus, which in combination with the recurrence with attention neurons causes long-lasting activation.

## Results

**Experiment 1**. In Experiment 1, 25 participants monitored two changing color disks, one on either side of fixation (Fig. 1). Between 2700 and 5400 ms into the trial a cue (a white or black ring, counterbalanced across participants) appeared for 27 ms around one of disks while a distractor ring (white or black, depending) appeared around the other disk. In the Smooth condition, the disks' color changed 16° around a circular color trajectory with an SOA of 108 ms. Smooth trials were intermixed within block with Random condition trials where the color changed pseudo-randomly (each color was at least 30° away from the previously presented color), also with an SOA of 108 ms. In both conditions, the disks continued to change for 810 ms after the cue. At the end of each trial, participants were asked to report the color of the disk at the time of the cue by moving their mouse horizontally to update a test disk along the presented color trajectory and clicking when the color best matched memory. The minimum distance between the reported color and the stream of presented colors was used to calculate the serial position error (SPE) and then multiplied by the SOA for a selection latency in time. For all experiments, a permutation analysis was conducted in order to test whether the selection latency of two conditions was significantly different and 95% confidence intervals were generated through bootstrap sampling. A within-subjects Cohen's *d* is reported for all comparisons[12].

When colors changed smoothly, the mean selection latency was 113 ms, 95% CI = [95, 131] (Fig. 2). However when the colors changed randomly, the mean selection latency was 2 ms, 95% CI = [−22, 28]. This reduction in selection latency across conditions

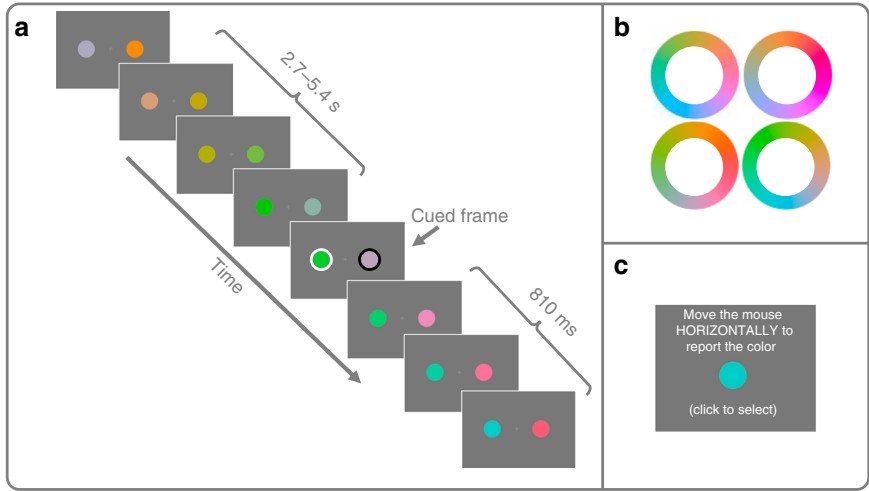

**Fig. 1 Experimental paradigm for Experiments 1 and 2. a** Participants fixated on a cross in the middle of the screen while two disks changed color (see "Methods" for details). A cue (white or black ring) flashed around one disk during the stream as a distractor disk flashed around the other for 27 ms. The disks then continued to change until the end of the trial. Here, sample frames of the Smooth condition are depicted. **b** Four possible color rings. Two rings (one for each disk) was selected randomly at the start of each trial. **c** At the end of the trial participants report the color of the disk at the time of the cue by moving the mouse horizontally to make the test disk smoothly change color. Participants click the mouse to select the color that best matches memory.

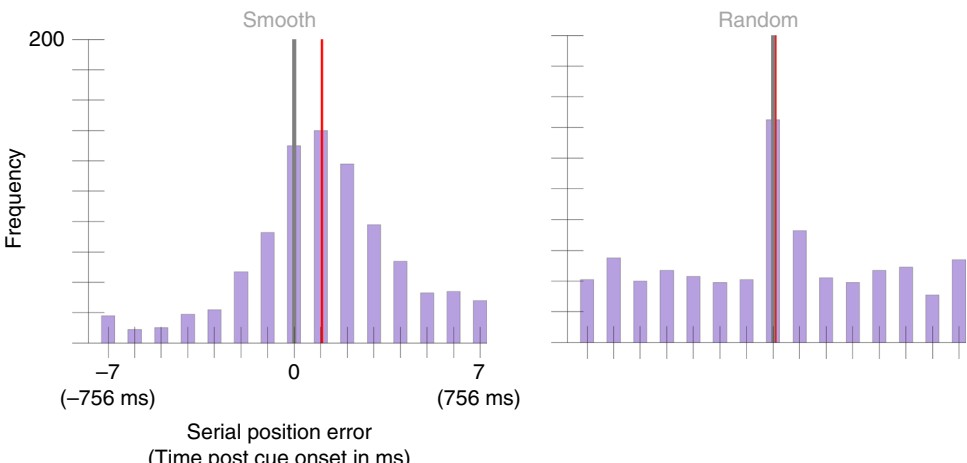

**Fig. 2 Serial position error (also translated into time) histograms aggregated across 25 participants in Experiment 1's Smooth and Random conditions.** Error has been discretized by bucketing trials based on the presented color that was closest to the reported color (see Results section). The gray vertical line indicates the position of the cued color (position zero). The red vertical line marks the condition mean (113 ms, 95% CI = [95, 131] for the Smooth condition and 2 ms, 95% CI = [−22, 28]). A two-sided permutation analysis showed a significant difference between conditions, $p < 0.001$, $d = 1.21$.

was significantly different from zero, $p < 0.001$, $d = 1.21$. This experiment was replicated in a second, independent sample of 25 participants. In this second sample, the Smooth condition had a mean selection latency of 150 ms, 95% CI = [128, 163] and the Random condition had a mean selection latency of 34 ms, 95% CI = [8, 59] (Fig. 3). Again, this difference was significant, $p < 0.001$, $d = 0.94$. There is an obvious difference in the variance between the Smooth and Random distributions, but because color errors and time are conflated in the Smooth condition but not in the Random condition, this difference is hard to interpret. For example, Gaussian noise added to the color perceived or reported in the Smooth condition would result in a Gaussian distribution of serial position errors, whereas in the Random condition it would contribute to the uniform distribution. Future work could test whether another effect of the visual transients presented in the Random condition is to encourage more precise temporal binding.

In Experiment 1 and its replication the Random condition yielded a reduced selection latency compared to the Smooth condition. These results suggest that something about the smooth changes leads to a later selected feature value than when stimuli are jumping randomly through feature space (i.e. the Random condition).

One possible explanation of the longer latency is that the smoothness of the stimulus changes extended attentional engagement. If attention is indeed dragged along by a stimulus' temporal autocorrelation, then increasing the correlation between colors from one time point to the next should further increase the duration of attentional drag. To test this, in Experiment 2 we compared the Smooth condition of Experiment 1 to a condition where the colors on successive frames are more similar to one another.

**Experiment 2.** Experiment 2 included both the Smooth condition of Experiment 1 and a condition where the color trajectory was

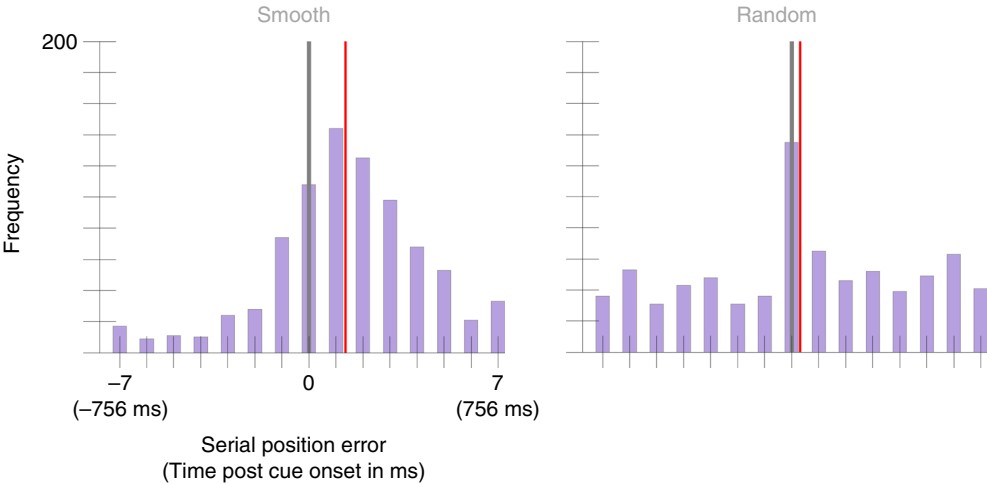

**Fig. 3 Serial position error histograms aggregated across 25 participants Experiment 1's replication study.** The gray vertical line marks the cued color position and the red vertical line marks the condition mean (150 ms, 95% CI = [128, 163] in the Smooth condition and 34 ms, 95% CI = [8,59] for the Random condition). A two-sided permutation analysis showed a significant difference between conditions, $p < 0.001$, $d = 0.94$.

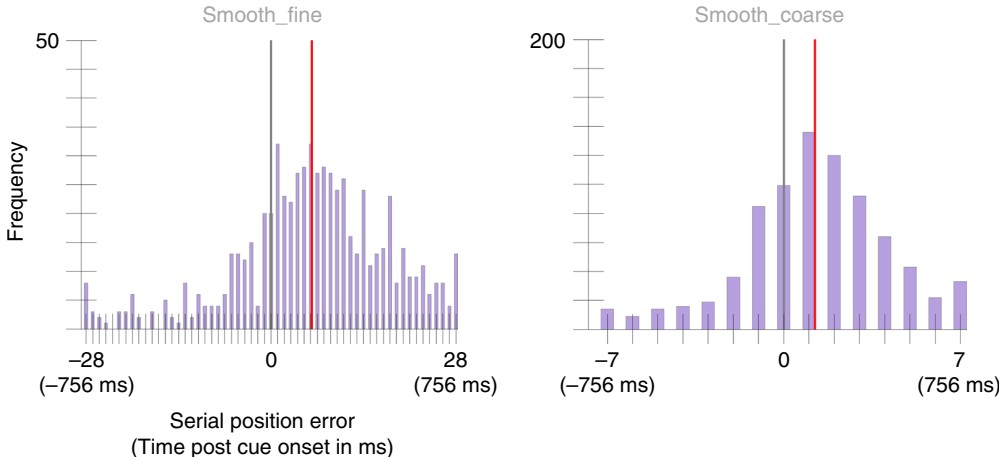

**Fig. 4 Serial position error histograms aggregated across 25 participants from Experiment 2's Smooth_fine and Smooth_coarse conditions.** Note there are four times the number of positions in the Smooth_fine condition as there are in the Smooth_coarse condition for the same window of time since colors updated in the Smooth_fine condition at four times the rate of that in the Smooth_coarse. The gray vertical line marks the position of the cued color. The red vertical line marks the condition mean (166 ms, 95% CI = [147, 185] in the Smooth_fine condition and 133 ms, 95% CI = [116, 150] in the Smooth_coarse condition. A two-sided permutation analysis indicates a significant difference between conditions, $p = 0.03$, $d = 1.45$.

even smoother. In this experiment the disks changed 16° along the color trajectory every 108 ms in the Smooth_coarse condition (identical to the Smooth condition of Experiment 1) or 4° every 27 ms in the Smooth_fine condition. Importantly, both conditions presented the same rate of change along the circular color trajectory, only the granularity of change was increased from the Smooth_coarse to the Smooth_fine condition. The attentional drag theory predicts the Smooth_fine condition will result in a longer latency of color selection.

The mean selection latency was 166 ms, 95% CI = [147, 185] for the Smooth_fine condition and 133 ms, 95% CI = [116, 150] selection latency for the Smooth_coarse condition (Fig. 4). The mean selection latencies were significantly different across conditions, $p = 0.03$, $d = 1.45$. A second, independent sample of 25 participants was run on the same paradigm to replicate these results and found a selection latency of 166 ms, 95% CI = [149, 182] in the Smooth_fine condition and a selection latency of 109 ms, 95% CI = [94, 123] in the Smooth_coarse condition (Fig. 5).

Again, there was a significant difference between conditions, $p < 0.001$, $d = 1.72$.

The results of Experiment 2 showed a significant reduction in the average selection latency from the Smooth_coarse to Smooth_fine presentation and this finding was replicated. Together, these results support the hypothesis that smooth feature changes leads to prolonged attentional engagement and causes a later selection of feature information.

However an alternative explanation for the combined results of Experiments 1 and 2 is that instead of the increased disruption of smoothness from Smooth_fine to Smooth_coarse to Random presentation, it is actually the decrease in predictability across conditions that is producing the selection latency reduction. The increased difference in color space from one update to the next in the Smooth_coarse condition compared to the Smooth_fine condition may make it more difficult for the visual system to extrapolate the next color. If the visual system is sampling from a generated internal representation instead of the current stimulus

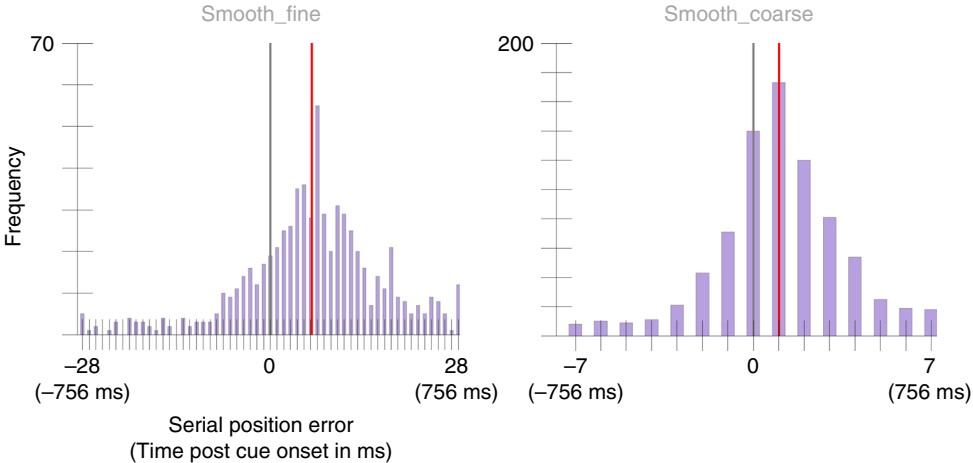

**Fig. 5 Serial position error histograms aggregated across 25 participants from Experiment 2's replication study.** The gray vertical line marks the position of the cued color. The red vertical line marks the condition mean (166 ms, 95% CI = [149, 182] in the Smooth_fine condition and 109 ms, 95% CI = [94, 123] in the Smooth_coarse condition). A two-sided permutation analysis showed a significant difference between conditions, $p < 0.001$, $d = 1.72$.

input, this increased difficulty might reduce how far into the future the system can predict and result in what appears to be an earlier selection latency. This effect would manifest similarly in comparing Smooth to Random selection in Experiment 1, as the Random condition does not offer a predictable change in color through time.

It would be seem counterintuitive for prediction effects to be the cause of these latency differences, since predictability is typically assumed to reduce errors rather than increase them. Regardless, to test the influence of predictability in causing these latency effects, Experiment 3 showed disks that changed in either a Smooth or Random pattern prior to the cue and switched to other pattern after the cue to test whether smoothness before or after the cue influenced selection latency.

**Experiment 3**. Experiment 3 tested whether the selection latency differences seen in the previous two experiments were a result of the visual system generating predictions of the color trajectory. The theory would be that in generating these predictions, when the cue triggers attention, the color sampled is from the internal representation (a future value) which leads to an appearance of selecting information later in time. To test this, 20 undergraduates participated in an experiment with a similar paradigm to the previous two used. However this time, in the Smooth-to-Random condition, the colors of the disks changed smoothly prior to the cue and randomly afterwards (in accordance with the smooth and pseudo-random color change of Experiment 1). These trials were intermixed within in block with Random-to Smooth trials where the disks changed randomly before the cue and smoothly afterwards. The internal model theory would predict that when the color changes are smooth before the cue, error distributions will be shifted forward along the before-cue trajectory as color reports are based on predictions. Note, that in this scenario, the color trajectory that would have followed after the cue was never actually presented to the participant. Conversely, when participants are first presented with randomly changing colors and then a smooth change after the cue, this theory predicts less of a shift in the error distribution as the system is unable to generate predictions prior to the onset of attention.

To test this, we measured the distance between the reported value and smooth trajectory, as it would have appeared if the presentation style was constant throughout the trial. The mean selection latency was 3 ms, 95% CI = [−16, 23] in the Smooth-to-

Random condition and 91 ms, 95% CI = [75, 106] in the Random-to-Smooth condition (Fig. 6). This difference was significant, $p < 0.001$, $d = 0.93$. Note that this pattern of results is opposite to that expected from the prediction theory. A replication study was also performed with a second sample of 30 participants, yielding a mean selection latency in the Smooth-to-Random condition was −6 ms, 95% CI = [−24, 11] and 47 ms, 95% CI = [33, 61] in the Random-to-Smooth condition (Fig. 7), significantly different, $p < 0.001$, $d = 0.68$. The data from both of these samples is presented in its raw form in the Supplementary information (Supplementary Figs. 1, 2).

The results from Experiment 3 and its replication support the idea that the later selection latency seen in the Smooth condition compared to the Random condition in Experiments 1 is not caused by the smoothness presented prior to the cue (as is assumed by prediction-based theories) but rather by the smoothness presented after the cue. However these results do not rule out the role of prediction entirely but rather one way it might have been used. This will be discussed further in the Discussion.

**Experiment 4**. As a further test of the effect of predictability of the color trajectory, in Experiment 4, the color trajectory was identical in every trial, with the exact same color trajectory for each disk. The only parameters that changed between trials then was the cue latency and the disk it was presented around. This is in contrast to Experiments 1, 2, and 3, in which the disk color trajectories were chosen at random from four different circles and direction of change was also randomly selected on each trial. If predictability leads to a later selection latency, this more predictable trajectory should produce an even later selection in this experiment than was observed in the Smooth_fine condition of Experiment 2.

The selection latency was 140 ms, 95% CI = [125, 154], 26 ms earlier than that of the Smooth_fine condition in Experiment 2 and its replication study (Fig. 8). As the hypothesis was that the predictable trajectory would lead to a later selection latency, which was not the case given that, if anything, it is earlier, no analysis was performed on this data. In this experiment, it was much easier to learn the color trajectory and yet the delay was not increased relative to the Smooth_fine condition of Experiments 2, reinforcing the conclusion from Experiment 3 that predictability of the sequence is not the cause of the latency to report a color.

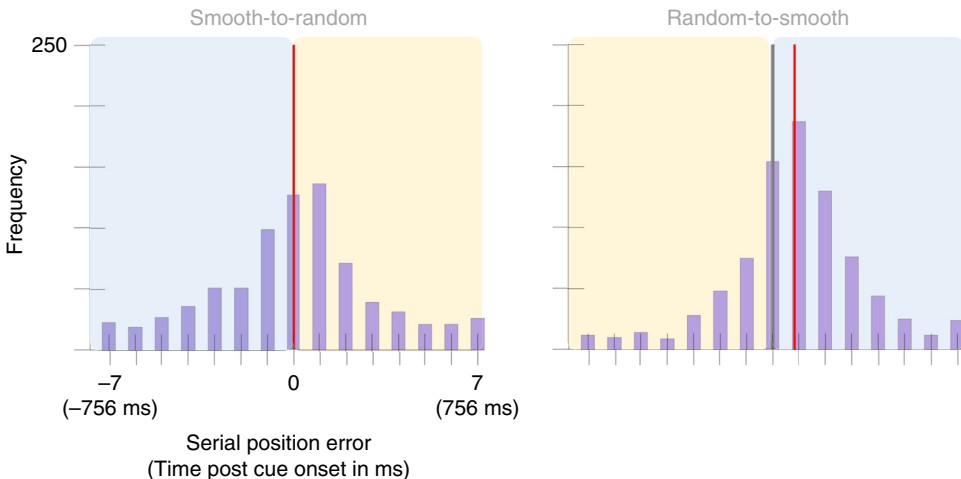

**Fig. 6 Serial position error histograms aggregated across 20 participants from Experiment 3.** The *x*-axis represents serial position error had the color been presented as smoothly changing both before and after the cue. For the half of the graph highlighted in blue, this trajectory matched what the participant was shown. Highlighted in yellow is the error compared to a presentation that was not shown to participants but was extrapolated by continuing the smooth trajectory presented. For each condition the gray vertical line marks the position of the cued color. The red vertical line marks the condition mean (3 ms, 95% CI = [−16, 23] in the Smooth-to-Random condition and 91 ms, 95% CI = [75, 106] in the Random-to-Smooth condition). A two-sided permutation analysis indicates a significant difference between conditions, *p* < 0.001, *d* = 0.93.

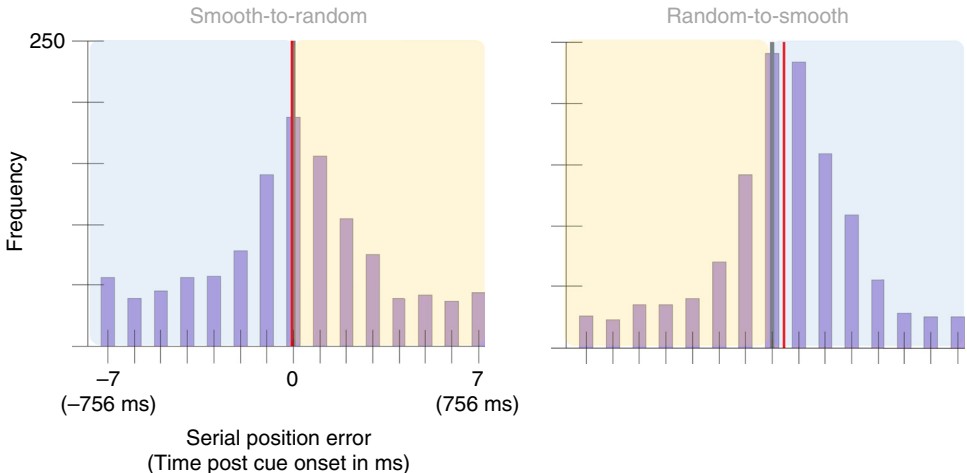

**Fig. 7 Serial position error histograms aggregated across 30 participants from Experiment 3's replication study.** As in Experiment 3's graph, the *x*-axis represents serial position error had the color been presented as smoothly changing both before and after the cue. The blue box highlights the portion of the trajectory that was shown to the participant. The yellow box highlights the portion of the trajectory that was not shown to participants but was extrapolated by continuing the smooth trajectory presented. The gray vertical line marks the position of the cued color and the red vertical line marks the condition mean in both graphs (−6 ms, 95% CI = [−24, 11] in the Smooth-to-Random condition and 47 ms, 95% CI = [33, 61] in the Random-to-Smooth condition). A two-sided permutation analysis showed a significant difference between conditions, *p* < 0.001, *d* = 0.68.

## Discussion

The experiments presented here found that the more smoothly the stimulus color changed, the later the reported color value was. This was only true, however, if the smooth change occurred after the cue onset. Like most empirical results, these findings have several potential explanations. We highlight our attentional drag theory, in which the smooth change of a feature keeps attention engaged for longer than a condition containing random transitions in feature space. While there are other possible mechanisms that could create such an effect, this explanation has the virtue of simplicity, and has been formally defined with a neurocomputational model. With this model, the attentional drag theory provides an explanation for why visual transients are such an effective visual stimulus, not just for capturing attention, but also for disengaging it.

The luminance transients that accompany sudden visual changes have long been suggested to capture or engage attention[13], and sometimes to disengage attention or control binding[11,14,15]. Such findings have typically been explained by proposals that transients directly affect attention – for example, that temporally high-pass cells have a particularly strong connection to neurons that elicit orienting[16]. We do not contest such theories here. However, our model does explain how the disengagement of attention may be a result not of transients per se but rather because transients are highly associated with the autocorrelation of features that results in lasting featural activity in our model. Future work may be able to tease these concepts apart, perhaps by superposing luminance transients on an otherwise smoothly changing stimulus.

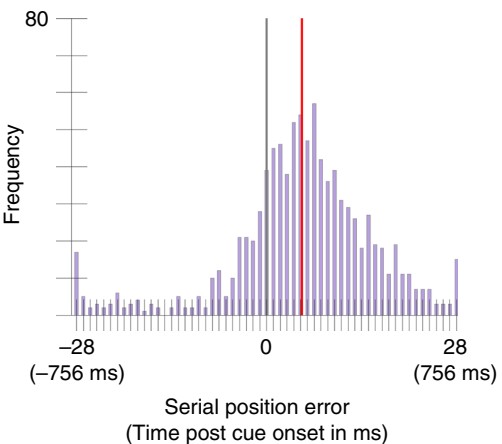

**Fig. 8 Serial position error distribution (translated into time as well) aggregated across 17 participants in Experiment 4.** The gray line indicates the position of the cued color (position zero) and the red line indicates the mean of the distribution (140 ms, 95% CI = [125, 154]).

Our core finding was shown by Experiments 1 and 2, revealing a relationship between selection latency and feature change, while Experiments 3 and 4 ruled out the possibility that this later selection was the product of future prediction based on a predictable trajectory presented prior to the cue. A possible alternative explanation for Experiment 3's results is that color in the smooth presentation are easier to perceive and so reports are biased toward colors presented in the smooth portion of the trial, rather than attention being dragged along in time, as proposed here. However such an explanation would not account for the increased latency in Experiment 1's Smooth condition where colors are presented smoothly throughout the trial and therefore should produce no bias. The attentional drag hypothesis provides a single explanation for the entire set of observed results.

To explain how the temporal engagement of attention could be extended by presenting a sequence of smooth feature changes, a computational model was developed. The model is based on the idea that attention is a recurrent process, as described in a previous model of attentional fluctuations over time (i.e. eSTST[17]). The eSTST model posits that attention is part of a positive feedback loop in which incoming sense data (e.g. the cue) triggers the deployment of attention at a given spatial location. This attention amplifies feedforward processing at that location to facilitate the encoding of information into memory and, consequently, amplifies its own input from that spatial location. As described in previous studies[17], with typical RSVP stimuli this feedback loop leads to brief attentional episodes on the order of 150 ms in duration, which explains the prevalence of effects like lag-1 sparing in the attentional blink. The new model proposed here elaborates on the eSTST model's basic architecture by incorporating biologically plausible tuning curves in the input neurons responsive to a given feature (e.g. color, orientation). For the sake of simplicity, this model is a simplification of eSTST in that it does not include memory encoding processes and runs at a finer resolution in time. This simplification is necessary because the eSTST model simulates the encoding of discrete stimuli as used in standard RSVP experiments. Future work will incorporate innovations, such as a distributed binding pool[18] that can encode stimulus values selected from a continuum.

The attentional drag model here has three components for each spatial location: an array of color-sensitive neurons, a cue sensitive neuron, and an attention neuron (Fig. 9). To simulate neuronal membrane potential, a set of differential equations[19] is used as a simple abstraction of excitation and leak current. To

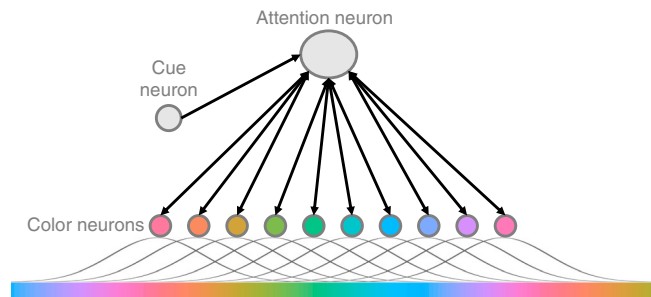

**Fig. 9 A schematic of the model used to explain the attentional drag theory.** Black arrows indicate excitatory connections. For illustration purposes only 10 color neurons are depicted but in the model there are 360. Each neuron is receives input from presented colors under a Gaussian distribution. When the cue neuron is triggered it excites the attention neuron over threshold which triggers the excitatory feedback loop between the attention neuron and the color neurons.

understand the drag phenomenon in more detail, consider that because each color-sensitive responds to an array of colors according to its tuning curve[20], the presentation of a single color will activate a distributed population of neurons. Thus, in the model color is represented as a vector representing the activation of 360 stimulus-responsive neurons with Gaussian-shaped tuning curves, which very roughly approximate V1 cells[20]. Therefore, any one color evokes a gradient of activation across the population of 360 neurons. A consequence is that when similar colors are presented sequentially (as in the Smooth condition) overlapping neural representations are activated sequentially, resulting in individual neurons being activated for longer and reaching higher levels of activation (Fig. 10a). Conversely, when colors are presented in random order, the color presented at one time step is less likely to have overlapping activation with the color presented in the previous and following time steps, resulting in less accumulation over time (Fig. 10b). In summary, smooth feature changes result in greater activation across the population of sensory neurons.

This greater activation of sensory neurons as a result of smooth color changes is not enough, according to the model, to explain the longer sampling latency observed in the data. The recurrent feedback in the model between attention and sensory neurons was required. It is widely accepted that reporting a stimulus involves feedback from higher-order neurons to sensory neurons[21], which in our model is accomplished by the "attention neuron". The attention neuron is triggered by a cue-responsive neuron. Specifically, when the cue is presented, this cue neuron is excited over threshold and sends excitation to the attention neuron. Once the attention neuron has been excited over threshold, it amplifies the excitation being received by all of the color neurons. For color neurons which are currently not receiving excitation (i.e. the color they are responsive to is not being presented) this attentional amplification does nothing. But for colors that are responsive to the current input, this attentional amplification excites them over threshold, allowing them to excite the attention neuron in turn. Thus, the presentation of the cue triggers the deployment of attention and consequently puts the system into an excitatory feedback loop. The establishment and maintenance of the feedback loop is how the model instantiates the engagement of attention.

In the case of randomly presented colors, when the overall activation of color neurons is lower, this loop has a short duration, decaying quickly after the offset of the cue (Fig. 10d). However when the system is presented with a smoothly changing color stimulus and color neuron activation accumulates from one

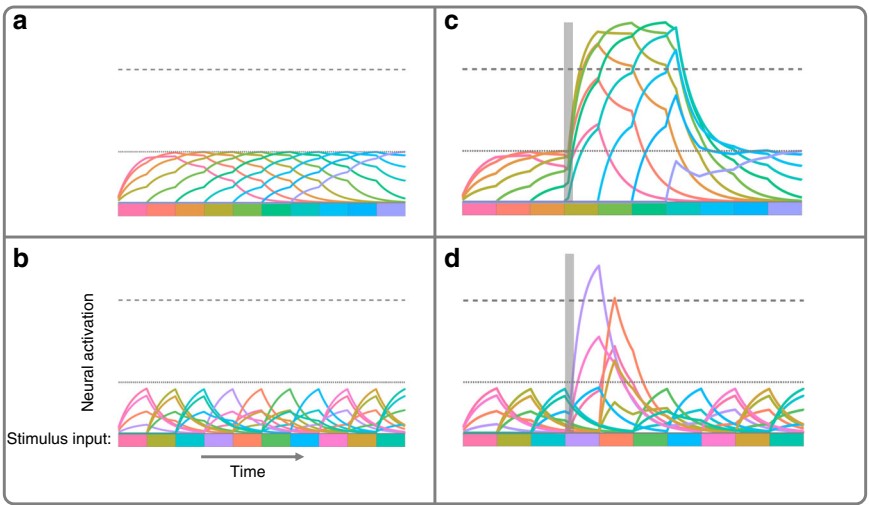

**Fig. 10 Model simulations of color neuron activation.** In each panel the lower dotted line marks the maximum activation that color neurons reach without attentional amplification when smooth color change is inputted into the system and the upper dashed line marks the color neuron threshold that, when crossed, allows color neurons to send forward excitation to the attention neuron. The bottom bar of colors in each panel show the sequence of colors presented to the system in time. The difference in the color bars and lines have been exaggerated from the real stimulus input for the purposes of visualization. The left column shows neural activation without the presence of attentional amplification when color are changing smoothly (**a**) or randomly (**b**). Importantly, overall neural activation is less in the random case compared to the smooth. The graphs in the right column show neural activation when the cue neuron triggers the deployment of attentional amplification and the attentional excitatory feedback loop is established. When colors are changing smoothly (**c**) and there is a higher level of activation of the color neurons, this attractor state is maintained for longer and later presented colors are excited over threshold. When colors are changing randomly (**d**) this attractor state is only maintained briefly, resulting in the cued color being excited over threshold and no others. In both **c** and **d** the transparent, vertical, gray bar denotes the duration of the cue presentation.

frame to the next, this greater overall activation of the color neurons provides more excitation to the activated attention neuron. Therefore, when the cue neuron triggers the deployment of attention, this feedback loop is sustained for longer in the Smooth condition compared to the Random one (Fig. 10c). This extended attentional engagement results in a longer epoch of excited colors in the Smooth condition. An additional experiment, included in the Supplement, supports the model by showing this phenomenon is robust to different cue durations (Supplementary Notes).

This model does not make specific claims regarding how the color is selected for memory encoding and subsequent report, but presumes that whatever that process is, the extended attentional engagement that means greater activation for later-occurring colors leads to a higher probability of reporting a later color in the Smooth condition compared to the Random condition. Additionally, it may be that selecting a color from the extended color representation in the Smooth condition requires an endogenous form of attention that has slower response times[3] and is poor at selecting information with a short latency from a dynamic stimulus[11].

While future work is needed to understand the ultimate selection process, the model does simulate the timing of the attentional window. Taking the midpoint of that window as a proxy for the selected color, the model yields a difference of 107 ms between the Smooth and Random conditions (similar to the 111 ms and 116 ms difference between conditions observed in Experiment 1 and its replication study). In comparing Smooth_fine to Smooth_coarse, the model simulates a 55-ms difference (a 33-ms and 75-ms difference between conditions was observed in Experiment 2 and its replication study). Without specifying a method of color selection, the model provides a formalized account of the attentional drag theory here and simulates the pattern of results seen in the empirical data. This general architecture could easily be incorporated into other models of attention, such as eSTST.

A crucial point to emphasize is that our model does not make any functional distinction between whether smoothness extends attentional engagement at a location or if a transient in color disengages attention, as here smoothness and the presence of a transient were perfectly confounded, as typically they are: lack of smoothness implies a discontinuity. Finally, a basic property of the model is that the increase in selection latency results from the degree of smoothness presented after the cue. Experiment 3 provided evidence that this is indeed the critical variable, thus ruling out an alternative account based on sampling from predictions of an internal model.

This is an abductive model in that it accounts for the empirical results with a simple set of neural mechanisms that are plausible inasmuch as they extend a model that has had success with other attentional phenomena[17]. However, previous research has had a different interpretation of similar findings. Namely, Sheth et al.[7] similarly found a long selection latency when reporting the color of a disk at the time of a cue as the disks smoothly changes between red and green. Sheth and colleagues proposed that, as the color changes smoothly, priming occurs, creating a ramp-up in activation of subsequent colors along the trajectory. When attention is triggered by the cue, attention resets this build-up of activation and so the selection of color is delayed until priming can once again ramp-up a color's activation sufficiently to be selected. This accounts for their results as well as the selection latency of the Smooth condition presented here. However, this theory, as is, would not explain the pattern of selection seen in the Random condition. Explaining this additional finding requires the priming theory to add some sort of immediate enhancement performed by attention in order to privilege the cued color for selection. Without this additional form of selection, every color presented in the Random condition would have an equal chance of being selected. This attentional enhancement could possibly be incorporated into these alternative theories, in addition to the resetting of priming by attentional shifts. However, since this would require two distinct roles for attention in Sheth and

colleagues' theory, it could be argued that the attentional drag theory offers a simpler account for the experimental results. Another potential alternative theory could be that attention integrates information over a fixed interval in time after the cue. While this theory offers a simpler account than the attentional drag, our simulations suggest that an attentional window of fixed duration applied to both random and smoothly presented color changes is unable to account for the results seen here (Supplementary Fig. 7). Again, the experimental work here does not necessarily eliminate these alternative theories. However the results do suggest that such theories would require additional mechanisms in order to account for the same data points the attentional drag is able to simulate. Further elaboration of such alternatives, ideally with a computational instantiation would be a key step in testing the distinction between these accounts since it is difficult to compare models with several different components without a concrete specification.

The empirical results presented here support an attentional drag theory wherein gradual or smooth changes at a location maintain attentional engagement compared to a salient feature change which disengages attention. There are of course many contributing factors to selection latency, such as whether attention needs to shift from a previously engaged location[10] and the saliency of the cue which triggers deployment. However, what these results demonstrate is how feature dynamics affect sensory information extraction. When features make jumps in feature space, attention is able to select information at the time of the cue. This is consistent with previous findings that have shown a transient signal at a location aids in temporal sampling[11,22]. Conversely, when features change smoothly in time, selection is delayed. These results suggest that the visual system samples information differently depending on the nature of change of the input. As features often change smoothly in our visual world, understanding this phenomenon could provide important insights into how attention tracks stimuli. This functionality may provide an inherent form of event segmentation that combines visual information from a sequence of time points only when they are likely to comprise the same visual event, which may be complementary to the brain's parsing of event information on much larger timescales[23,24]. Similarly, these mechanisms may underlie how we construct object representations and help to explain why the visual system updates with new information in some circumstances and initiates a new episode or "object file" in others[25].

## Methods

**Experiment 1.** Twenty-five undergraduates from the Pennsylvania State University subject pool filled out informed consent forms before participating in this study. Participants were between 18 and 23 years old and had normal or corrected to normal vision. A second, independent sample of 25 participants were collected for the replication study. This experiment, and the subsequent experiments, were approved by the Pennsylvania State University Internal Review Board.

Stimuli were presented using a 16-inch CRT monitor (1024 × 768, 75 Hz refresh rate) placed 63.5 cm away from participants' headrest, using Psychtoolbox functions[26]. The stimuli for this experiment consisted of two colored dots, presented on either side of a fixation cross (3° of visual angle separation between the center of each dot and the fixation). Each dot subtended 2.7° of visual angle. Four color rings (sets of colors that each form a closed path through color space) were generated for this experiment. The values of each ring were calculated by first setting the a, b coordinates (in L*a*b color space) for each ring's center ([0, 0], [50, 0], [20, 50], [−40, 30]). The radius of each ring was set to 60 and the L parameter for each ring was set to 70. Given these parameters, 360 points were calculated along each ring (Fig. 1b). All stimuli and paradigm code, as well as data and analysis code for this and the following experiments is available on the OSF [https://osf.io/hujwb/]. The monitor was not color-calibrated and thus the actual L*a*b values differ, but the property that the colors formed a closed smooth path was preserved. The starting color of each disk as well as the direction of change through the color ring was randomly selected at the start of each trial. Thus there were eight color trajectories to reduce the participants' ability to predict the color that would appear, although this was shown not to matter in Experiment 4.

Participants were asked to keep their eyes on a fixation cross in the middle of the screen while monitoring a changing color disk on both sides. To decrease the predictability of the cue location as well as to mimic previous literature which has used dual RSVP streams[9], all the experiments here used two disks. Participants were told to remember the color of the disk when the cue (a white or black ring, counterbalanced across participants) flashed around it. Between 2700 and 5400 ms after the stimuli first appeared, the cue ring simultaneously onset around one of the disks and a distractor ring (the opposite color of the cue, white or black depending on the participant) appeared around the other disk for 27 ms. The timing of the cue, and the side on which the cue appeared, was randomized across trials. The two disks continued to change for another 810 ms after the offset of cue and distractor ring (Fig. 1).

At the end of every trial a report screen instructed participants to move the mouse horizontally to change the color of a test disk to match that of the cued disk as it was at the time of the cue. The test disk was presented at the center of the screen. As the participants moved the mouse, the test disk changed colors through the cued disk's color ring at a rate of 2° per pixel. When the participant found the color that best matched their memory, he or she clicked the mouse to select it. Afterwards participants were given a score from 0 to 100 in which a score of 100 indicated that they had reported the exact color presented at the time of the cue and score of 0 meant that they had reported the color 180° away from the target color. The score did not indicate the direction of error.

There were two conditions in Experiment 1, each with 44 trials that were intermixed within block. The first condition was the *Smooth* condition where the color of the disks advanced 16° every 108 ms through the selected color ring. The second condition was the *Random* condition where the color disks changed pseudo-randomly every 108 ms. The selection of the next color was randomly chosen from the color ring of that disk with the constraint that the new color had to be at least 30° away from the previous color. Thus, in both conditions color information changed with the same presentation rate but the relationship in color space across time points differed.

The serial position error was calculated the same way for all experiments. On each trial the difference was taken between the reported color and the cued color as well as every color presented in the seven positions before and after the cue. The serial position error then was determined by the minimum difference (i.e. the color which was closest to the reported value) and translated into milliseconds using the SOA. The trial was excluded if the reported value was an equal distance from the two closest colors presented (i.e. a tie). Additionally, if this minimum difference between the reported color and the color of the closest matching stimulus was >16° of the 360° of the color circle, that trial was excluded from analysis. The rationale was that such reports are especially likely to be guesses or gross misperceptions of the color. This second exclusion criteria was chosen a priori as a compromise between the Random condition, which requires no such criteria, and the Smooth condition which does in order to avoid edge effects, where the histograms bins at the edges of the minimum color response distribution (i.e. ±7 serial positions) would be assigned any guess responses from other portions of the color wheel. To check for robustness, additional analyses were performed varying the threshold for both Experiments 1 and its replication. The results are included in the Supplementary information (Supplementary Table 1) with corresponding figures (Supplementary Figs. 3, 4).

In Experiment 1 these exclusion criteria resulted in an average of 5 trials (SD = 3) excluded from the Smooth condition and 7 trials (SD = 2) excluded from the Random condition, per participant. In the replication study, 5 trials (SD = 3) on average were excluded from the Smooth condition and 8 trials (SD = 3) in the Random condition per participant.

**Experiment 2.** Twenty-five undergraduates volunteered from the Pennsylvania State University subject pool to participate in this study. All participants were between 18 and 23 years old with normal or corrected to normal vision. Informed consent was obtained for each participant prior to the study in accordance with the IRB office of Penn State University. A second, independent sample of 25 participants was run as a replication study. The same stimuli used in Experiment 1 were used in Experiment 2.

A procedure similar to Experiment 1's was used in Experiment 2. Participants again monitored two color disks, one on either side of fixation, and were asked to report the color of the disk at the time a cue ring flashed around it. The same cueing and report method used in Experiment 1 were used here. Instead of Smooth and Random, the two conditions were *Smooth_coarse* and *Smooth_fine*. The Smooth_coarse condition here is the same as the Smooth condition of Experiment 1 where colors advanced 16° every 108 ms along the color ring. In the Smooth_fine condition the color of the disks advanced along the color ring by 4° every 27 ms. Importantly, in both conditions the average rate of change through color space was the same but finer time steps were taken in the Smooth_fine condition. As in Experiment 1, there were 44 trials per condition, intermixed within block.

As there were four times as many colors presented in the Smooth_fine condition as the Smooth_coarse, the number of positions (or color values presented) in the analysis before and after the cue was 28 in the Smooth_fine condition as opposed to 7 in the Smooth_Coarse. Similarly, since the Smooth_fine condition presents colors closer to one another along the color ring, the analysis for both conditions was restricted to only trials where the minimum distance between

a reported color and presented colors was 8° in this experiment. This criterion was chosen as a compromise in order to apply the same trial exclusion procedure to both conditions.

With these exclusion criteria, an average of 6 trials (SD = 3) from the Smooth_coarse condition and 12 trials (SD = 3) from the Smooth_fine condition were excluded from analysis in Experiment 2. In the replication study, per participant, an average of 6 trials (SD = 3) from the Smooth_coarse condition and 12 trials (SD = 3) from the Smooth_fine condition were excluded from analysis. A second analysis, included in the Supplementary information, lowers the criterion to 4° (ideal for the Smooth_fine condition). This naturally results in the exclusion of a greater number of trials, but it yields the same findings in both Experiment 2 and its replication study (Supplementary Figs. 5, 6).

**Experiment 3**. A sample of 20 undergraduates age 18–23 with normal to correct to normal vision were used for this experiment and a second sample of 30 participants were used for the replication study with the same recruitment and consent procedure as outlined in Experiments 1 and 2. The same stimuli used in Experiments 1 and 2 were used in this experiment.

In this experiment, participants again maintained fixation on a cross in the middle of the screen and monitored two changing colored disks. Participants were told to report the color of the disk at the time of the cue. The same cueing and reporting method used in the previous two experiments were used here. In Experiment 3 there were two conditions. In the *Smooth-to-Random* condition the color disks changed color in accordance with the Smooth presentation style used in Experiment 1 (16° along the color ring every 108 ms). At the time of the cue, the presentation style changed to random where colors were pseudo-randomly presented every 108 ms (with the same constraints as outlined in Experiment 1). In the *Random-to-Smooth* condition the color disks changed as in the Random condition of Experiment 1. At the time of the cue the disks began to change in the Smooth presentation style, beginning the color trajectory from the color presented at the time of the cue.

As in Experiment 1, trials were excluded if their minimum distance from the color presented in the cue position or those in the 7 positions before or after the cue exceeded 16° of if the minimum distance was equal between two presented colors. These exclusion criteria led to, on average, 11 trials (SD = 4) from the Smooth-to-Random condition and 7 trials (SD = 4) from the Random-to-Smooth condition being excluded per participant. In the replication study 12 trials (SD = 5) were excluded from analysis for the Smooth-to-Random condition on average while 9 trials (SD = 5) were excluded on average from the Random-to-Smooth condition.

**Experiment 4**. Seventeen undergraduates, age 18–23 with normal or corrected to normal vision, were used for this experiment with the same recruitment and consent procedure as outlined in Experiments 1, 2, and 3. The same stimuli used in Experiments 1, 2, and 3 were used in this experiment. However this time only one color ring, of the four used in the previous studies, was used here. This color ring was calculated by setting the a, b coordinates in L*a*b color space to [0,0] and the L parameter to 70. Centered at this position in color space, 360 equidistant points around a circle with a radius of 60 was calculated.

A similar procedure was used in Experiment 4 as was used in Experiment 2. Again, participants fixated on a cross in the middle of the screen while monitoring two changing color disks. This time, both disks used the same color ring trajectory. At the start of every trial the disk on the left started its trajectory at the 1° (RGB values: [255, 120.28, 172.2]) color while the disk on the right started at the 180° color (RGB values: [0, 198.1, 168.8]). Both disks moved through their trajectories in the same direction on every trial. The side and time of the cue was randomized as in the previous experiments. As in the Smooth_fine condition of Experiment 2, the colors updated at a rate of 4° every 27 ms. The reporting method used was identical to that in the previous experiments. There was only one condition in this experiment with a total of 88 trials. As in Experiment 2 the 8° criterion was used for excluding trials due to the similarity of colors presented from one time point to the next. This resulted in an average of 29 trials excluded per participant (SD = 12).

**Analysis**. For all experiments presented here, a permutation analysis was conducted in order to test whether the selection latency of two conditions was significantly different. For instance, in Experiment 1, on every permutation, participants' Smooth and Random condition position errors were randomly shuffled into two bins. The average position error of these two bins was then computed and the difference between those averages was recorded. This was repeated ten thousand times, building a null distribution of mean differences that could have resulted if there were no difference between the two conditions. The real mean difference between the Smooth and Random condition was then compared to this permuted null distribution. The percentile location of the real mean difference in the permuted null distribution is the *p*-value. This analysis was also repeated with permutations at the subject level. Here the mean difference between conditions for each subject was calculated. There was a 0.5 chance for each subject that this difference was calculated by subtracting the mean of Smooth trials from the mean of Random trials or vice versa. The average of across all of these subject differences was then calculated. This process was repeated 10,000 time to build a distribution of average condition differences, generated under the assumption that

there is no true difference between conditions (the null distribution). The true grand average was then compared to this distribution in order to calculate a *p*-value. The dichotomous outcomes of all tests throughout the paper were identical when using this method and the *p*-values were of similar magnitude.

Lastly, in order to calculate confidence intervals around each condition's mean selection latency, ten thousand bootstrapped samples were drawn, and the mean was calculated. This bootstrapped distribution of mean selection latency then allowed us to calculate the 5th and 95th percentiles.

**Statistics and reproducibility**. Experiments 1–3 were all replicated once using an independent second sample of participants. Experiment 1 had 25 participants and its replication had 25 participants. Experiment 2 had 25 participants and its replication had 25 participants. Experiment 3 had 20 participants and its replication had 30 participants. There was no replication of Experiment 4.

**Reporting summary**. Further information on research design is available in the Nature Research Reporting Summary linked to this article.

## Data availability
The datasets generated and analyzed during the current study are available on the Open Science Framework (OSF) [https://osf.io/hujwb/]. These datasets were used to generate Figs. 2–8. The code for generating these figure is also included in the same OSF project. A reporting summary for this Article is available as a Supplementary Information file.

## Code availability
For all experiments presented, the stimuli, code for running the paradigms, and the code for running all of the analysis and generating the figures of this paper have been uploaded to the Open Science Framework (OSF) [https://osf.io/hujwb/].

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

## Acknowledgements

This work was funded by NSF Grant BCS-1331073. Special thanks to Mackenzie Corcoran, Natalie Crow, Kyrie Murawski, and Aubrey Spak for their help with data collection.

## Author contributions

A.O.H. contributed to the motivation of the study, the relation to other literature, the interpretation, and to the revision. C.C.F. and B.W. developed the experimental paradigms, the analysis method, and the computational instantiation of the attentional drag model. The paper was written primarily by C.C.F. with assistance from B.W.; Coding of the model and parameter fitting was done by C.C.F.

## Competing interests

The authors declare no competing interests.
