## [Peer Review File · Nature Communications]

Reviewers' comments:

Reviewer #1 (Remarks to the Author):

The current manuscript investigates how people respond to attentional cues when dealing with continuously changing stimuli. Previous work has found more "lag" for smoothly changing stimuli. The current work again finds that temporal autocorrelation of the stimuli induces more lag -- and shows this occurs particularly for smooth changes *after* the cue. The manuscript also presents a model whereby attention is engaged longer when features are changing smoothly than when it is not, resulting in this lag, and giving some new insight into how attentional allocation might work in a more realistic scenario where objects do not suddenly appear and disappear but linger and change.

Altogether, I think this is a nice set of results, demonstrating how to unify the previous findings that have sometimes found a lag in selection and providing new empirical insights (e.g., the lag depending on post-cue, rather than pre-cue smooth changes). The authors have a model-based theory for how to interpret these results, arguing that attention "drags" (aka, accumulates information from sensory cortex for longer) in the smooth conditions. Thinking about these results in terms of overlapping color channels seems extremely useful and I think many insights can arise from thinking this way and by computationally instantiating models of this framework. However, I did have a concern, which is that I found myself wondering whether there might be a different interpretation of the results that does not depend on any change in how long attention is engaged at the location, but simply the way the chosen color is recreated from the color channel activity.

In particular, if I imagine that each stimulus evokes activity across color channels, and then this activity decays with time, and I try to reconstruct the cued color from stimulus activity 150ms post-cue (e.g., the time my brain manages to figure out to check low-level visual cortex to ask what color is present), this intuitively seems sufficient -- on its own -- to predict the current effects, without any change in attention's durations or allocation (e.g., with no "drag"). I'm imagining just taking the population activity at 150ms post cue and assuming a maximum-likelihood population decoding rule (the maximum likelihood color to have caused the current population activity, as people like Bays et al 2014 use in models of color memory).

Imagine color i is the cued color in the smooth condition. First we present color $i-10$, which evoke activity in color $i-10$ as well as in color i (because of the broad tuning of the neurons), and then this activity begins to decay. Then we present color i . Presenting this color causes a significant amount of activity in color channel $i+10$, the next color that will be shown (because of the broad tuning of the neurons), which again begins to decay. Then, color $i+10$ is shown. There is now more activity in the color channel $i+10$ than there is in the random condition (say, color j), and more activity in color $i+10$ than $i-10$ (because of decay). Now, we try to decode what color happened ~ 150 ms ago. We use a maximum likelihood read-out that knows about our own neural tuning and decay the current color channel activity at exactly this moment. Don't we end up with a bias "forward" in time (because activity caused by the presentation of stimulus $i-10$ has less influence vs. the one at $i+10$) that is greater in smooth than

random, and primarily dependent on the smoothness after the cue, not before?

I guess, rephrasing this: Can't we argue for two simple principles rather than argue for attentional drag? (1) it is far more ambiguous what set of activity to attribute to the "cued" timepoint vs. uncued timepoints in the smooth condition because of broad tuning, and (2) because newer stimuli have stronger activity than older stimuli (due to decay), there is a bias forward, not backward, caused primarily by post-cue stimuli?

Maybe this account is not that different from what you are proposing -- but it seems to me it has no real role for "attentional drag" in it, since everything is about what stimulus activity looks like at a fixed point in time. Possibly contrary to the drag account, as far as I can tell this "moment-in-time" account predicts that almost the entire effect should occur from just presenting a single "similar" color after the cue, since that is what is causing the confusion in the color channels at the moment people are trying to access low-level visual activity. That is, a completely "random" condition but where the color right after the cued color happens to be similar to the cued color (you may even have these trials?) ought to cause almost the exact same effect, even though there would be no perception of smooth color changes and no real role for "drag" of attention beyond simply confusing the i and $i+10$ item.

Reviewer #2 (Remarks to the Author):

Review of Callahan-Flintoft, Holcombe & Wyble, "A delay in sampling information from temporally autocorrelated visual stimuli".

This is a very interesting series of experiments contrasting perception of auto-correlated versus non-correlated stimuli, showing differences, and suggesting that this might be due to differences in attention. The behavioral methods are sound, the results clearly described, and the addition of a computational model compelling. I particularly like the inclusion of replications for every experiment! I nevertheless have some issues, and in particular an alternative hypothesis that I am not sure the current data can rule out.

Alternative hypothesis.

If the perceived color of any stimulus (smooth or random) is the result of averaging sensory samples from a short time-window, responses should be pulled towards future stimuli more in the smooth than in the random condition, as in the present results. This explanation does not require any input from "attention", or for processing to be gated by the type of stimulus.

If in generating a color response, subjects average all color samples in a time-window (for example, of 100 ms) that begins the moment the cue is presented, then on a smooth trial, the difference between the response and the color presented at the moment of the cue should be biased towards subsequent stimuli. In the random condition, this should not be the case since the average difference between the

color at cue onset and each of the subsequent colors is zero.

However, the sampling from a time-window explanation predicts no difference between smooth-coarse and smooth-fine, since all that changes between coarse and fine conditions is the color step size, and not the overall extent of color change. Therefore, to be convinced that this experiment rules out the simple averaging over a time-window explanation, I would like to be convinced of the following (which I could not really find in the manuscript under its current form): that the difference between the conditions emerges even in color space. Looking at Figures 4 and 5, it appears that the average position error is 6 in the fine condition, which corresponds roughly to 24° on the color wheel. Compared to the coarse condition (experiments 1, 1b, 2 and 2b) which looks like 1 to 1.5 steps (corresponding to 16 to 24° on the color wheel).

Secondary issues (in the order of the manuscript).

Methods.

Why are there 2 colored disks? It should be possible to measure the same effects with a single disk per trial. Having two disks forces participants to split attention, which might interact with the way in which attention subsequently samples from one of the disks. The authors might check this by looking at whether there is a relationship between the two disks: are responses pulled towards the color presented in the opposite (non-cued) disk? If so, is this different between the two conditions (random versus smooth)?

What was the initial color of the test disk, and were responses biased towards it?

Results.

In all experiments, the authors report only the difference between conditions (for example, random versus smooth), but I would like to know if the lag in each condition was different from zero.

The Figures present serial position error in number of color steps. To translate this into something meaningful, one has to remember the size of the steps (for example, 16° in the Smooth-coarse or 4° in the Smooth-fine) and recalculate. The results present differences in time, for which one has to translate from steps to color to time. The Figures would be much more interpretable if serial position were translated into color error, and both that and time were reported in the text.

Experiment 3. Figures 7 and 9 show data for which there is no hypothesis and no analysis. I suggest the authors remove them.

General Discussion.

The order of information is not straightforward. For example, Figure 11 is mentioned after Figure 13. The computational model should be able to make some quantitative predictions, which could then be compared with the data. I suggest the authors do just that, rather than keep the model purely descriptive.

Figures.

In the pdf version of the article, Figures 4 and 5 are superimposed.

To be able to see the legends of some of the figures, I had to resize them manually in the Word

document.

Reviewer #3 (Remarks to the Author):

Dear Editor,

Thank you for the opportunity to review the manuscript by Callahan-Flinthoff et al.

The manuscript described a set of experiments studying the phenomenon that when a smoothly varying stimulus visual property (such as the position of a moving object) is instantaneously sampled, a delay is usually observed, whereas this delay is reduced or eliminated when the visual stimulus is characterised by transients (for example successive flashes). The authors advocate a model in which the temporal autocorrelation of the stimulus causes “attentional drag”, and outline a computational model for how this would work in the colour domain.

The topic and question are clearly interesting and a model that could explain the substantial body of data in the literature would be very valuable. Furthermore, I am generally sympathetic to the idea of “attentional drag” as it is proposed here: that transients help the attentional system to disengage, whereas ongoing stimulus cause it to “stick” and therefore yield a later sample.

However, I feel that the manuscript needs to do more to persuade me that we have made any progress in understanding this. The background provided by the manuscript is too limited, and to my mind the experiments only superficially address the issue. I outline some specific points below.

1.

There is a large body of available literature about the delays inherent in sampling from streams of information in response to a cue. In particular, Reeves and Sperling (1986) also used paradigms with cued RSVP streams to identify attentional sampling latency. In a paradigm quite similar to the current paradigm, Weichselgartner and Sperling (1987) used single rsvp streams with a luminance cue, and identified TWO peaks in the resulting latency distribution.

It is striking that the data reported here show essentially one peak at zero latency, whereas previous literature has found some small but measurable delays using RSVP streams (which should be conceptually equivalent to the discontinuous colour condition here). This needs to be addressed.

2.

I feel the discussion of the data is unsatisfying. At multiple points in the paper, two histograms are compared (smooth and discontinuous), and the only conclusion that is extracted is the mean difference. However, the much more striking difference is the qualitative difference in the shape of the distribution: there is a normal distribution of reported colour in the smooth condition, but a single isolated peak (and a high baseline of noise responses) in the discontinuous case. I feel this is a far more salient feature of the results than the subtle shift in mean. At the very least, this forms a problem that cannot be ignored

when calculating the mean, as the distribution shapes are different (in any case, the high baseline noise in the discontinuous condition would pull the mean towards zero).

3.

It also makes me wonder what part of the variability in response is attributable to temporal inaccuracy, and what part is due to colour memory imprecision. I imagine it should be possible to get an indication of this by analysing the responses to the discontinuous condition. At any rate, this should be considered in the manuscript.

4.

Finally, and this could be considered a problem or a feature of the current model, but the synchronous discontinuities, which are much more salient in the random than smooth condition, seem to encourage temporal binding in a way that the smoothly changing stream does not. This also seems evident in the data: the Smooth distribution is normally distributed, whereas the Random condition seems more All-or-None for the correct feature.

Minor point: on page 13, the 95% confidence intervals (8-34) exclude the actual mean (75 ms).

Minor point: I would reverse the order of presentation of Figures 6 and 7. Figure 7 shows what was actually observed, which to me makes more sense to read and digest before reorganising the data to show what would have been but wasn't.

We really appreciate the reviewers' efforts and feel that this manuscript has greatly improved based on their input. Below, please find our response to each of their points. Thank you again for your help with this work.

Sincerely,

Chloe Callahan-Flintoft, Alex O. Holcombe, Brad Wyble

Reviewer #1

1) I did have a concern, which is that I found myself wondering whether there might be a different interpretation of the results that does not depend on any change in how long attention is engaged at the location, but simply the way the chosen color is recreated from the color channel activity.

In particular, if I imagine that each stimulus evokes activity across color channels, and then this activity decays with time, and I try to reconstruct the cued color from stimulus activity 150ms post-cue (e.g., the time my brain manages to figure out to check low-level visual cortex to ask what color is present), this intuitively seems sufficient -- on its own -- to predict the current effects, without any change in attention's durations or allocation (e.g., with no "drag"). I'm imagining just taking the population activity at 150ms post cue and assuming a maximum-likelihood population decoding rule (the maximum likelihood color to have caused the current population activity, as people like Bays et al 2014 use in models of color memory).

Imagine color i is the cued color in the smooth condition. First we present color $i-10$, which evoke activity in color $i-10$ as well as in color i (because of the broad tuning of the neurons), and then this activity begins to decay. Then we present color i . Presenting this color causes a significant amount of activity in color channel $i+10$, the next color that will be shown (because of the broad tuning of the neurons), which again begins to decay. Then, color $i+10$ is shown. There is now more activity in the color channel $i+10$ than there is in the random condition (say, color j), and more activity in color $i+10$ than $i-10$ (because of decay). Now, we try to decode what color happened ~ 150 ms ago. We use a maximum likelihood read-out that knows about our own neural tuning and decay the current color channel activity at exactly this moment. Don't we end up with a bias "forward" in time (because activity caused by the presentation of stimulus $i-10$ has less influence vs. the one at $i+10$)

that is greater in smooth than random, and primarily dependent on the smoothness after the cue, not before?

I guess, rephrasing this: Can't we argue for two simple principles rather than argue for attentional drag? (1) it is far more ambiguous what set of activity to attribute to the "cued" timepoint vs. uncued timepoints in the smooth condition because of broad tuning, and (2) because newer stimuli have stronger activity than older stimuli (due to decay), there is a bias forward, not backward, caused primarily by post-cue stimuli?

Maybe this account is not that different from what you are proposing -- but it seems to me it has

no real role for "attentional drag" in it, since everything is about what stimulus activity looks like at a fixed point in time. Possibly contrary to the drag account, as far as I can tell this "moment-in-time" account predicts that almost the entire effect should occur from just presenting a single "similar" color after the cue, since that is what is causing the confusion in the color channels at the moment people are trying to access low-level visual activity. That is, a completely "random" condition but where the color right after the cued color happens to be similar to the cued color (you may even have these trials?) ought to cause almost the exact same effect, even though there would be no perception of smooth color changes and no real role for "drag" of attention beyond simply confusing the i and $i+10$ item.

Thank you for bringing up this point. This is actually an alternative that we have discussed amongst ourselves quite a lot and your feedback really helps to identify the points we still need to clarify in describing our model. Here we outline why an attentional drag is necessary to produce the complete set of behavioral results. To this end, text has also been added to the manuscript as well (page 20). Additionally, in formulating a response to your comments we developed some new figures that we think are much better illustrations of these key points we're trying to make and so have substituted those in for some of the original figures that may have been harder to interpret.

If we assume that activation of the color neurons in the model is accumulating all throughout the stream, then there isn't much of a forward bias in time because just as the cued color (i) activates the next color's neuron ($i+1$), it also activates the previous color's neuron ($i-1$). In Figure 1 here we have plotted the activation of the cued color neuron and its two neighboring color neurons (the color differences have been exaggerated for clarity). Observe that at the moment the cue is presented, the $i-1$ color is the most active. Data from human observers shows that people are most likely to report the cued color in the random condition so in the model we set a time window for sampling the population where that outcome is most likely (similar to the hard coded 150 ms you proposed). To test this alternative model, if we apply that same sampling window to the smooth case in the absence of attentional drag, there is no forward bias (see Figure 1 top).

However, there is a forward bias in activation for the Smooth condition if we assume that color activation only starts accumulating once the cue is presented, since earlier colors then are not allowed the same amount of time to build activation. Figure 2 plots the activation of the cued color and the two preceding colors in this scenario. As you can see later colors reach a higher

Figure 1: Neural activation for the cued color and the color presented just before and after for the Smooth (top) and Random (bottom) conditions. The color boxes at the bottom of each graph show the stream of presented colors (108 ms each). The transparent patch marks the presentation of the cue (27 ms). The two vertical grey lines mark the time window in which the cued color (i.e. purple) has the maximum activation in the random condition in these simulations. The time points of these lines are calculated from the random activation and then applied to both the random and smooth condition. The horizontal dashed line marks the max activation of neurons during smooth presentation. The differences in presented colors and the line color have been exaggerated compared to how they actually appeared in the experiment for visual clarity.

level of activation. If the system selects the color of greatest activation this would create a forward shift in reports without incorporating an attentional drag. However without any attentional amplification, all of the colors in the Random condition have equal activation which would predict that reports would be uniformly distributed among colors presented after the cue. The data clearly shows however a bias for the cued color over the other colors in the stream for the Random condition. The benefit of the attractor state that the attentional drag implements is that it amplifies the cued color in the random case above the other colors in the stream as well as creating enhancing activation of later colors in the stream for the smooth case (see new figure 10 in the paper).

To your point about presenting just one “similar” color after the cued color in an otherwise random condition, we agree that this would induce a drag as well. We’ve taken this “similar” idea to be something like taking two frames out of the Smooth condition by presenting the cued color and then a color 16° along the color ring. The attentional drag predicts that the build-up in activation caused by presenting two colors sequentially that activate overlapping populations of neurons would extend attentional engagement for longer than if the subsequent color activated a

Figure 2: Neural activation for the cued color and the 2 subsequent colors in the Smooth (top) and Random (bottom) conditions, under the assumption that color activation only begins building after the onset of the cue. The color boxes at the bottom of each graph represent the colors presented to the system (108 ms each). The transparent, grey patch indicates the presentation of the cue (27 ms). The dotted, horizontal line marks the max amplification reached by color neurons in the Smooth condition.

more disparate population of neurons. By extension the theory would support your assumption that there should be more of a lag in selection in the Random condition if the color after the cued color were randomly similar to the cued color. Unfortunately the current dataset does not allow for that analysis as the Random condition was explicitly designed to create transients by setting

the sequence to be pseudo-random such that colors were at least 30° away from the previous color.

Reviewer #2

1) If the perceived color of any stimulus (smooth or random) is the result of averaging sensory samples from a short time-window, responses should be pulled towards future stimuli more in the smooth than in the random condition, as in the present results. This explanation does not require any input from “attention”, or for processing to be gated by the type of stimulus. If in generating a color response, subjects average all color samples in a time-window (for example, of 100 ms) that begins the moment the cue is presented, then on a smooth trial, the difference between the response and the color presented at the moment of the cue should be biased towards subsequent stimuli. In the random condition, this should not be the case since the average difference between the color at cue onset and each of the subsequent colors is zero. However, the sampling from a time-window explanation predicts no difference between smooth-coarse and smooth-fine, since all that changes between coarse and fine conditions is the color step size, and not the overall extent of color change. Therefore, to be convinced that this experiment rules out the simple averaging over a time-window explanation, I would like to be convinced of the following (which I could not really find in the manuscript under its current form): that the difference between the conditions emerges even in color space. Looking at Figures 4 and 5, it appears that the average position error is 6 in the fine condition, which corresponds roughly to 24° on the color wheel. Compared to the coarse condition (experiments 1, 1b, 2 and 2b) which looks like 1 to 1.5 steps (corresponding to 16 to 24° on the color wheel).

Figure 3: Serial position error histograms for Experiment 1 where color reports were replaced with the circular mean of the cued color and 2 preceding colors. Noise from a Gaussian distribution (mean = 0, standard deviation = 12) was added to each report in order to better simulate a distribution. The grey vertical line marks the position of the cued color. The dip in report of the cue +2 position (as well as the smaller dip in the cue-1 and cue+4 positions) is due to the fact that each presented color was sampled under the requirement that it be at least 30 degrees away on the color ring from the previously presented color.

Thanks for pointing out this alternative. For argument sake we would need to expand the averaging window out a bit as 100 ms would only allow for the inclusion of one color in the Random and Smooth condition (SOA of 108 ms) and therefore predict that in both conditions participants select the cued color. To see what the results pattern would be like under this alternative theory we reanalyzed the data substituting the actual color reported by the participant with the circular average (as these are values along a color ring) of the cued color and the next two presented colors in the sequence for the Smooth and Random condition of Experiment 1 (Figure 3). In order to simulate a distribution for visualization purposes, noise from a Gaussian distribution with mean = 0 and standard deviation = 12 was added to each report. As you proposed, this method does create the forward shift we see in the real data of the Smooth condition. However the pattern simulated in the Random condition does not match that seen in the actual data as the cue+1 and cue+2 positions are frequently sampled.

Another issue with this alternative hypothesis, as you point out, is that it would predict the same degree of forward bias between the Smooth_fine and Smooth_coarse condition of Experiments 2 and 2b, as any window of fixed duration in time would cover the same area of color space in both conditions. Figure 4 illustrates this. As the overall rate of change along the color ring was the same in both conditions, a difference in time necessarily equates to a difference in color space. For example, in Experiment 2, Smooth_coarse had a mean selection latency of 133 ms while Smooth_fine had a mean selection latency of 166 ms. This translates to a 20 degree shift in color space for the coarse condition and 25 degree shift in the fine condition. We agree with your later point about including the individual condition means for clarity. This is addressed further below (point 4).

Figure 4: Example color change for the Smooth_Fine (top) and Smooth_Coarse (bottom) conditions over a 648 ms window. The black lines mark the mean selection latency for the respective condition.

2) Why are there 2 colored disks? It should be possible to measure the same effects with a single disk per trial. Having two disks forces participants to split attention, which might interact with the way in which attention subsequently samples from one of the disks. The authors might check this by looking at whether there is a relationship between the two disks: are responses pulled towards the color presented in the opposite (non-cued) disk? If so, is this different between the two conditions (random versus smooth)?

We decided to use two disks to decrease the predictability of the location of the cue prior to its onset. In this way we hoped to limit the effects of more endogenous forms of attention and focus on reflexive mechanisms. The two disk design also more closely mirrors the dual RSVP experiment (Goodbourn & Holcombe, 2015) whose results we were trying to explain. We have added some justification of the experimental design in the methods section (page 22) to clarify this. In unpublished data, Ludowici & Holcombe have found that the fewer the number (of letter streams), the earlier the letters reported (lower latency), and they find evidence that this is due to monitoring the stimuli prior to the cue coming on. You are correct though that we have also seen increased selection latency when only one stimuli is smoothly changing on the screen.

It's difficult to test whether reports were biased towards the other stream as the two disks moved along different color trajectories and the participant was restricted to the target trajectory during reporting. In Experiment 4 the color trajectories were the same across both disks and so in Figure 5 below we have replotted the data comparing reports to the un-cued stream. Here the grey vertical line marks the position of the distractor ring, which was presented simultaneously with

the cue. In this experiment the two streams were always 180 degrees apart (explaining the large peak at -28, which is 112 degrees away) and so there are systematic differences here that are not easily pulled apart. This is definitely an interesting question but requires a specific experiment designed to examine reports biased by the distractor stream or in fact swap errors. With two rapid streams of letters, Goodbourn & Holcombe (2015) did a similar analysis and found that the number of reports of stimuli from the non-cued stream was negligible.

Figure 5: Serial position error of Experiment 4's data this time compared to the un-cued instead of the cued stream. The grey vertical line marks the position at which the cue appeared (though in this stream it was the distractor ring that appeared).

3) What was the initial color of the test disk, and were responses biased towards it?

The initial color of both disks was randomly chosen at the start of each trial (as was the trajectory of each disk and the direction along which it changed). Additional language has been added to the manuscript to clarify this point (page 22). After consideration, we decided not to analyze this effect to avoid making an excessively long paper.

4) In all experiments, the authors report only the difference between conditions (for example, random versus smooth), but I would like to know if the lag in each condition was different from zero.

Thank you for this point. Originally we did not want to make strong claims about the exact time of selection but rather just that selection was delayed in some conditions relative to others. However we agree with your point that including the actual condition latency would provide clarity to the reader and in fact make the work more relatable to previous studies which have reported absolute rather than relative selection latencies. This has been implemented in each of that results sections of the manuscript.

5) The Figures present serial position error in number of color steps. To translate this into something meaningful, one has to remember the size of the steps (for example, 16° in the Smooth-coarse or 4° in the Smooth-fine) and recalculate. The results present differences in time,

for which one has to translate from steps to color to time. The Figures would be much more interpretable if serial position were translated into color error, and both that and time were reported in the text.

We really appreciate your point here on how to make the results most intuitive to the reader and went back and forth quite a bit trying to figure out how to do this. We understand that it is cumbersome to translate serial position error (SPE) to time and/or color. To remedy this, we have put in translations of the x-axis into time.

Unfortunately it's not entirely clear how to report meaningful results using color degree. In the smoothly changing conditions, the color advances along its circular trajectory in time and so the two are conflated (e.g. a 108 ms lag translates to 16 degrees along the color wheel given the rate of change). However in the random case, color is not changing consistently so, for instance, a 108 ms lag could translate to a 30 degree color separation or 95 degree color separation depending on the trial. Because of this we have decided to leave the measurement units in time in order to make them comparable across conditions and experiments.

6) Experiment 3. Figures 7 and 9 show data for which there is no hypothesis and no analysis. I suggest the authors remove them.

Thanks for this feedback. These graphs have been removed.

7) The order of information is not straightforward. For example, Figure 11 is mentioned after Figure 13.

The discussion has been revised in order to progress in a more straightforward manner and in agreement with the order of the figures.

8) The computational model should be able to make some quantitative predictions, which could then be compared with the data. I suggest the authors do just that, rather than keep the model purely descriptive.

We're really happy that you (and the other reviewers) like the inclusion of the model. We specifically did not want to make strong claims about how the ultimate feature value is selected for report as there are a number of potential methods that could be employed and our data are not suggestive as to a specific one. Rather we wanted to demonstrate with the model that an extended attentional window is possible with smooth feature change and this allows the selection of later presented colors. However we can read out from the model the simulated duration of the attentional window and so using the midpoint of that window as a proxy for the selection latency we were able to calculate a delay in selection of 107 ms comparing the Smooth to the Random condition of Experiment 1 (similar to the 111 and 116 ms differences measured in Experiments 1

& 1b) and a delay in selection of 55 ms comparing the Smooth_fine condition to the Smooth_coarse condition (similar to the 33 and 75 ms differences observed in Experiments 2 & 2b). Text describing this has been added to the manuscript (page 19).

9) In the pdf version of the article, Figures 4 and 5 are superimposed.

To be able to see the legends of some of the figures, I had to resize them manually in the Word document.

Sorry about that. We have fixed this in the revised version.

Reviewer #3

1) There is a large body of available literature about the delays inherent in sampling from streams of information in response to a cue. In particular, Reeves and Sperling (1986) also used paradigms with cued RSVP streams to identify attentional sampling latency. In a paradigm quite similar to the current paradigm, Weichselgartner and Sperling (1987) used single rsvp streams with a luminance cue, and identified TWO peaks in the resulting latency distribution. It is striking that the data reported here show essentially one peak at zero latency, whereas previous literature has found some small but measurable delays using RSVP streams (which should be conceptually equivalent to the discontinuous colour condition here). This needs to be addressed.

Thank you so for the suggested resources. We have included discussion of these papers in the introduction (page 3-4) and discussion (page 21). In regards to the Weichselgartner & Sperling paper, in that task participants were asked to report the earliest 4 digits that they remembered appearing after the luminance cue. Because of this stipulation the two distinct peaks likely demonstrate the presence of an attentional blink (though this phenomenon had not been named at the time). The first peak, which the authors label the “first glance”, shows a zero or near zero latency in the selection of information in response to the cue presentation. The subsequent dip in reporting of the numbers that immediately followed the first is likely the product of encoding suppression triggered by the processing of the first target. As the attentional blink concludes, encoding resources become available for later presented digits, resulting in the second peak or “second glance” process identified in the Weichselgartner & Sperling’s work. As the participants in our study are only tasked with reporting the earliest color they see once the cue onsets, we would expect only one peak (indicative of that initial selection). However this is an interesting potential avenue for future work.

To address your concerns about the selection latency of the Random condition in our work and latencies measured in comparable RSVP measurements we have included the condition selection means which show a 2 ms selection latency for the Random condition in Experiment 1 and a 34 ms selection latency in the replication study (Experiment 1b). This range is similar to selection

latencies seen in the previous literature that we discuss (e.g. Goodbourne & Holcombe's ~25 ms). However there are of course many other factors that could influence the latency of selecting information from a changing stream of information, such as the criteria by which a target is selected (e.g. a low-level feature cue, categorical membership of the target...etc), in addition to the temporal autocorrelation of the information itself. We have added text to the manuscript to address this point (page 21)

2) I feel the discussion of the data is unsatisfying. At multiple points in the paper, two histograms are compared (smooth and discontinuous), and the only conclusion that is extracted is the mean difference. However, the much more striking difference is the qualitative difference in the shape of the distribution: there is a normal distribution of reported colour in the smooth condition, but a single isolated peak (and a high baseline of noise responses) in the discontinuous case. I feel this is a far more salient feature of the results than the subtle shift in mean. At the very least, this forms a problem that cannot be ignored when calculating the mean, as the distribution shapes are different (in any case, the high baseline noise in the discontinuous condition would pull the mean towards zero).

3) It also makes me wonder what part of the variability in response is attributable to temporal inaccuracy, and what part is due to colour memory imprecision. I imagine it should be possible to get an indication of this by analysing the responses to the discontinuous condition. At any rate, this should be considered in the manuscript.

As points 2 & 3 are similar we have elected to address them as one. We agree that the difference in the shape of the distributions for the Smooth and Random conditions is very interesting. However it is difficult to compare the two. As you perhaps alluded to with the mention of variability resulting from temporal inaccuracy, in the Smooth condition, time and color space are conflated as there is a constant rate of change along the color trajectory in time. As such, the difference might easily be explained by imprecision in color report masquerading as temporal errors, causing the smooth condition to be much more spread out in time. It is difficult to tease apart what portion of the variability is due to a temporal smearing in selecting a color in time, noise introduced in working memory storage, or sensorimotor noise at reporting. We agree that the variability seen in the Random case is more likely temporally based as the color space variability is restricted by the boundaries set in bucketing reports (described on page 23). However even if we were to subtract that from the variability in the Smooth condition, we would not know what portion of the remaining variability was the product of color based or temporal based variability (as temporal variability is presumably different across conditions).

For these reasons we limited ourselves to a discussion of the mean shift as color-based variability in the Smooth condition should be normally distributed around the selected color and therefore should not contribute to a temporal shift. We take your point about the uniform component in the Random condition biasing the selection latency toward zero however as the median and mode selection latency are similar to the mean we feel this is a fair way of measuring temporal shift.

4) Finally, and this could be considered a problem or a feature of the current model, but the

synchronous discontinuities, which are much more salient in the random than the smooth condition, seem to encourage temporal binding in a way that the smoothly changing stream does not. This also seems evident in the data: the Smooth distribution is normally distributed, whereas the Random condition seems more All-or-None for the correct feature.

Thank you for this point (although again, the normal distribution might be entirely a result of noise at the response choice or memory stage rather than being related to temporal binding). This is a similarity between the model and the data that we had not described. We have added some text on this in the discussion (page 7).

5) on page 13, the 95% confidence intervals (8-34) exclude the actual mean (75 ms).

Thanks for catching this typo. This has been fixed in the revised version.

6) I would reverse the order of presentation of Figures 6 and 7. Figure 7 shows what was actually observed, which to me makes more sense to read and digest before reorganising the data to show what would have been but wasn't.

As we were over the journal's limit for the number of figures to be included in the manuscript and Reviewer 2 had advised to drop the raw data figures from Experiment 3 & 3b, we decided to move these figures to the supplemental.

Reviewers' comments:

Reviewer #1 (Remarks to the Author):

Thanks to the authors for their very clear response and visualizations. It seems both myself and Reviewer 2 had similar worries that simple models that start averaging perceptual activity after the cue can account for the results, and I confess I still don't quite understand why this is not true, even in the current version – for example, it seems to me in the Figure 1 in the response letter, there is a lot more activation of the i-1 color in the random condition than the i+1 color during the relevant time window (integrating over the time window, it appears to be ~3x as much of the i-1 as the i+1). And in fact in every experiment, in the random condition subjects do appear slightly more likely to report the i+1 color than the i-1 color, so this window of integration of evidence could indeed be considerably shifted to the right and still fit the data in the random condition. (Thinking of this as a window during which high level areas are accumulating the information from these color channels).

Indeed, the time window of integration visualized in this figure would need to be a bit later even just to give equal activation to i-1 and i+1 in the random condition, and even more to fit the pattern where i+1 is slightly greater than i-1. To me, this still seems like it would completely explain the “drag” in the smooth condition without any ‘drag’ – people report the color with the strongest integrated signal over this ~100ms window that occurs after the cue would seem to be a sufficient model to explain the data in all conditions. But perhaps I'm still not understanding something critical, since the authors seem to believe they have strong evidence against this account. Clearly they interpret the “read-out” of these activations based on a model different than mine (I'm just integrating the amount of activation for each color in the relevant window and assuming that reflects how likely people would be to pick it).

To be clear, showing that smoothly changing and randomly changing stimuli differ in effective “lag” because of the nature of color channel activation and the need for selection seems perfectly interesting to me as an empirical claim, and the theoretical account of this seems useful. The empirical claim is well supported and interesting, and the paper provides strong and interesting theory as well. I just continue to feel the papers makes a pretty strong claim about “attentional drag” and I don't really see any reason to suppose the “integration time” must be, or even is, different in the two conditions.

Reviewer #2 (Remarks to the Author):

The authors have responded fully to my previous comments. No further comments.
Thérèse Collins

Reviewer #3 (Remarks to the Author):

The authors have improved the paper with their revisions, and I can see how they have addressed the concerns of the other reviewers particularly.

I still have concerns about the qualitative difference in the nature of the variability difference between conditions, but agree with the authors that it is essentially impossible to say something meaningful about that on the basis of the present data.

I don't think this problem is so severe as to invalidate the paper, and the authors now appropriately acknowledge this constraint.

Reviewers' comments:

Reviewer #1 (Remarks to the Author):

Thanks to the authors for their very clear response and visualizations. It seems both myself and Reviewer 2 had similar worries that simple models that start averaging perceptual activity after the cue can account for the results, and I confess I still don't quite understand why this is not true, even in the current version – for example, it seems to me in the Figure 1 in the response letter, there is a lot more activation of the $i-1$ color in the random condition than the $i+1$ color during the relevant time window (integrating over the time window, it appears to be $\sim 3x$ as much of the $i-1$ as the $i+1$). And in fact in every experiment, in the random condition subjects do appear slightly more likely to report the $i+1$ color than the $i-1$ color, so this window of integration of evidence could indeed be considerably shifted to the right and still fit the data in the random condition. (Thinking of this as a window during which high level areas are accumulating the information from these color channels).

Indeed, the time window of integration visualized in this figure would need to be a bit later even just to give equal activation to $i-1$ and $i+1$ in the random condition, and even more to fit the pattern where $i+1$ is slightly greater than $i-1$. To me, this still seems like it would completely explain the “drag” in the smooth condition without any ‘drag’ – people report the color with the strongest integrated signal over this $\sim 100ms$ window that occurs after the cue would seem to be a sufficient model to explain the data in all conditions. But perhaps I'm still not understanding something critical, since the authors seem to believe they have strong evidence against this account. Clearly they interpret the “read-out” of these activations based on a model different than mine (I'm just integrating the amount of activation for each color in the relevant window and assuming that reflects how likely people would be to pick it).

To be clear, showing that smoothly changing and randomly changing stimuli differ in effective “lag” because of the nature of color channel activation and the need for selection seems perfectly interesting to me as an empirical claim, and the theoretical account of this seems useful. The empirical claim is well supported and interesting, and the paper provides strong and interesting theory as well. I just continue to feel the papers makes a pretty strong claim about “attentional drag” and I don't really see any reason to suppose the “integration time” must be, or even is, different in the two conditions.

[RESPONSE] Thank you for your review. We really appreciate the amount of time and consideration you have given our paper. We agree that the integration model you propose here is a straightforward one and, as you and Reviewer 2 demonstrate, a popular alternative. Below we have outlined our argument for why this model, as you've stated it here, would not account for the complete pattern of results we obtained. In brief, while an integration model could account for the pattern in the smooth condition given particular assumptions about the time window, a straightforward version of this model cannot account for the smooth and random conditions.

We have made mention of this in the discussion and include this more in-depth discussion (including a figure) in the supplement.

To illustrate the predictions of the integration account, in Figure 1 of our original response letter (copied over to this letter for ease of discussion), we placed the 100 ms window such that the cued color in the Random condition had the greatest activation throughout the window and then therefore was the color most likely selected. We then applied that same window to the Smooth condition to show that the cued color in the Smooth condition was also the strongest in that window. Thus the straightforward integration account does not exhibit an increased latency for

Figure 1: Neural activation for the cued color and the color presented just before and after for the Smooth (top) and Random (bottom) conditions. The color boxes at the bottom of each graph show the stream of presented colors (108 ms each). The transparent patch marks the presentation of the cue (27 ms). The two vertical grey lines mark the time window in which the cued color (i.e. purple) has the maximum activation in the random condition in these simulations. The time points of these lines are calculated from the random activation and then applied to both the random and smooth condition. The horizontal dashed line marks the max activation of neurons during smooth presentation. The differences in presented colors and the line color have been exaggerated compared to how they actually appeared in the experiment for visual clarity.

the smooth condition, relative to the random one.

You are correct that this particular window could (and in fact should) be shifted later in order to simulate our finding that, in the Random condition, the cued color is the most likely selected and the +1 color is more likely selected than the -1 color. However, under the

integration theory you propose, this later window would not result in a later selection for the Smooth condition. To illustrate this, we took a 100 ms window starting at the onset of the cue and calculated the area under the curve (AUC) for each color neuron. We then slid this window later in 1 ms increments for 200 ms. For each of these 100 ms windows we calculated the AUC for each color neuron. Figure 2 is an illustration of this, where the y-axis in the bottom row graphs is the color neurons' AUC for a given 100 ms window and the x-axis is the window's starting time, post cue onset. As you supposed, in the original window used in Figure 1, the -1 color had twice the strength of the +1 color, which is not supported by the data.

Figure 2: Top: Color neuron activation for the Smooth (left) and Random (right) conditions (copied from Figure 1). Bottom: Area under the curve of the each color neuron's activation for a sliding integration window. Time 0 on the bottom graph indicates the integration of a 100 ms window starting at the time of cue onset. The same width window (100ms) was then slid later by 1 ms for 200 ms to show the change in AUC for each color neuron as a function of integration window placement. The black vertical line in each of the bottom graphs indicates the upper bound of the integration window placement (as sliding the window any later would result in the +1 color having a stronger activation than the cued color).

Using the observed response latency from the Random condition as a benchmark, we set an upper bound on how late the integration window could be by marking the point at which AUC of the +1 color neuron within the window would exceed the cued color neuron's AUC. Applying the same boundary to the Smooth condition, we see that no matter where the window is placed prior to that boundary, the cued color neuron, and not the +1 color neuron, shows the strongest activation. **In fact, there is no point along the entire continuum from 0-200ms in which the Smooth condition favors a later response than the Random condition.**

There of course may be other ways of implementing the integration theory to fit the data found in this paper and we have added an acknowledgement of this in the discussion section. However, using the formulation described by the reviewers it does not appear that an integration

window of fixed duration with a window that is shared between the random and smooth conditions can re-create the pattern of results we find.

****REVIEWERS' COMMENTS:**

Reviewer #1 (Remarks to the Author):

I appreciate the thoughtful response to my comments. The current manuscript will make a good addition to the literature.